# Landscape of Constitutional *SOX4* Variation in Human Disorders

**DOI:** 10.3390/genes15020158

**Published:** 2024-01-25

**Authors:** Mina Grippa, Claudio Graziano

**Affiliations:** 1SSD Genetica Medica, Dipartimento Materno Infantile, AOU Policlinico Modena, 41125 Modena, Italy; grippa.mina@aou.mo.it; 2Medical Genetics Unit, AUSL Romagna, 47522 Cesena, Italy

**Keywords:** *SOX4*, SOXopathy, intellectual disability, developmental delay, aortic aneurism, semicircular canals, hypomorphic variants

## Abstract

SOX proteins are transcription factors which play a role in regulating the development of progenitor cells and tissue differentiation. Twenty members are known, clustered in eight groups named A through H and sharing a common DNA-binding domain called the HMG (high-mobility-group) box. Eleven of the SOX genes have been associated with genetic disorders so far, covering a broad spectrum of developmental diseases. *SOX4* is a single-exon gene and belongs to the SOXC group, together with *SOX11* and *SOX12*. *SOX4* variants have been recently described to cause a highly penetrant but heterogeneous disorder, with a phenotypic spectrum ranging from mild developmental delays and learning difficulties to intellectual disabilities with congenital anomalies. Nineteen pathogenic variants have been reported to date, generally de novo, heterozygous, and inactivating, either stop–gain or missense, the latter ones primarily targeting the HMG domain. Further, a bi-allelic variant was reported in a single consanguineous family. Copy number variants leading to whole gene deletion or duplication are rare and not clearly associated with any neurodevelopmental disorder. Many open questions remain regarding the definition of variants of unknown significance, a possible role of missense variants outside the HMG domain, genotype–phenotype correlation, the range of phenotypic spectrum and modifying factors, and treatment options.

## 1. Introduction

Members of the SOX family of transcription factors are defined as based on the presence of a DNA-binding domain with homology to the high-mobility-group (HMG) box of *SRY* (sex-determining region Y) [1]. It is well recognized that the SOX family plays pivotal roles in many developmental and pathological processes, including male differentiation [2,3], eye development [4], skeletogenesis [5] and neurogenesis [6,7]. Twenty members are known, clustered in eight groups (SOXA to SOXH), and half of them are associated with human genetic disorders, termed “SOXopathies” [8].

*SOX4*, *SOX11*, and *SOX12* belong to the SOXC group. They are expressed in many progenitor cell types and have redundant roles in controlling cell survival and fate determination in response to various signalling pathways [9]. SOXC proteins have almost identical DNA-binding domains and show a high degree of conservation in their other known functional region, a transactivation domain located at the C terminus. Animal models first showed that *SOX4* and *SOX11* are essential developmental genes, since both *sox11*-null and *sox4*-null mice die in utero or at birth with multiple abnormalities [10,11]. *sox12*-null mice do not show any apparent phenotype, thanks to functional compensation by *sox4* and *sox11* [12]. Two de novo heterozygous missense variants in the *SOX11* HMG box have been linked to a human disorder characterized by intellectual disability (ID), growth deficiency, facial dysmorphism, and hypoplasia of the fifth digit [13]. The disease was classified as a “Coffin–Siris syndrome-like syndrome”. More de novo heterozygous mutations were later reported in patients with similar features, including *SOX11*-containing 2p25 deletions, a nonsense variant, and additional HMG-domain missense variants [14], but deeper phenotyping and analysis of DNA-methylation profiles proved that this condition is distinct from Coffin–Siris syndrome (CSS) [15]. Whole exome sequencing led to the identification of the first heterozygous *SOX4* mutations in 2019, when four de novo missense variants in the HMG box were reported in individuals with developmental delays and mild facial dysmorphism [16].

## 2. SOX4 Structure and Function

*SOX4* is a single-exon gene, and the open reading frame encodes a 474-amino acid protein, which includes two functionally important domains: namely, an HMG box and a C-terminal transactivation domain (TAD). The HMG box facilitates DNA binding, bending, and nuclear trafficking, whereas TAD mediates the interaction with different cofactors [17], although knowledge of critical residues within TAD is still missing. By binding to the minor groove of DNA, SOX4 alters chromatin architecture, leading to changes in transcriptional activities of downstream genes. SOX4 (and SOX11) has pleiotropic functions, which are likely to be mediated by distinct regulatory elements and downstream target genes involved in multiple developmental processes, including neurogenesis [18,19], heart development [10] and skeletal patterning [20], but also lymphocyte maturation [21] and more recently development of the inner ear [22]. The specific target genes of SOX4 have not yet been fully defined, but, among the genes that have been shown to be regulated by SOX4, some are important for neurodevelopment and disease (e.g., *RELN*, *DCX*, and *WDR45*) [19,23]. In the human brain, *SOX4* expression was found to be high in several regions (the dorsolateral prefrontal cortex, striatum, and cerebellar cortex) during the first two trimesters of embryonic gestation, and then to decrease progressively to reach a very low level by the 4th decade of postnatal life; the expression was higher in areas of active neurogenesis [16].

This widespread involvement in developmental processes is consistent with the heterogeneous set of anomalies which can be observed in individuals carrying *SOX4* pathogenic variants.

## 3. SOX4 Single Nucleotide Variants

### 3.1. Heterozygous Pathogenic Variants

In line with other SOXopathies, most *SOX4* causative variants reported to date are dominant, either loss of function or missense variants which target the HMG box [16,24,25]. Globally, 13 pathogenic missense and six stop–gain (one frameshift, five nonsense) variants have been described (Table 1 and Figure 1). These variants were all identified through exome or genome sequencing studies and were defined as pathogenic/likely pathogenic based on (a) in silico assessments (modification of highly conserved amino acids in the HMG box, in different vertebrates orthologos, and in other human SOX proteins; generation of premature stop codons; absence in gnomAD database (https://gnomad.broadinstitute.org/ accessed on 30 December 2023); predictions on the effects of missense variants on protein structure and function) and (b) functional assessments (weak or absent DNA binding and loss of transactivation activity in reporter gene assays).

Missense variants were defined as causative mainly based on their inability to bind DNA and activate transcription. Nonetheless, differences in functional tests were reported among distinct variants. For instance, most HMG variants were severely underrepresented in the nucleus when compared with SOX4 wild-type protein and did not bind DNA, whereas p.Leu99Pro was only slightly underrepresented, and p.Leu99Pro and p.Ala112Thr bound DNA weakly. Furthermore, p.Leu99Pro and p.Ala112Thr were weak in transactivation assays, whereas other HMG variants were fully inactive [24].

These differences may be a source of phenotype heterogeneity, but many more variants will need to be assessed to draw any conclusions.

Since *SOX4* is a single-exon gene, mutations that result in premature stop codons are expected to escape nonsense-mediated m-RNA decay (NMD) and produce a shorter, truncated protein, which may exert partial functions and/or impair functioning of the wild-type SOX4 copy through a dominant-negative effect. p.Gly44Argfs*2 and p.Glu27* variants fall very early in the open reading frame and encode a peptide which has an unknown function but lacks both functional DNA-binding and transactivation domains. The p.Tyr325*, p.Ser333*, and p.Ser347* variants removed 128–150 residues, including the functionally essential TAD, and p.Glu445* deprived TAD of its C-terminal segment. While the p.Gly44Argfs*2 peptide was undetectable in the cytoplasm and nuclei of transfected cells with SOX expression plasmids, p.Tyr325* and p.Glu445* were over-represented, especially in the nucleus, and p.Glu445*, although weak, was not fully inactive in the transactivation assay [24].

Pathogenic missense and truncating variants were also found to interfere with the activity of wild-type SOX4, suggesting a dominant-negative effect.

Parents were available to be tested in most (18/19) cases: 14 variants were de novo (4 stop–gain mutations, 10 missense), 4 transmitted (2 stop–gain, 2 missense). *SOX4* variants may thus be not fully penetrant, although it is not possible to draw definite conclusions. Only DNA samples from peripheral blood were tested; p.Gly44Argfs*2 was present with low-level somatic mosaicism in the patient’s mother, and this woman was reported to suffer neurocognitive issues. Also, the p.Asn64Lys variant was present in the patient’s father, a man with learning difficulties, mild facial dysmorphism, and limited extension of 5th finger. Two parents (the mother of a patient with p.His119Tyr and the father of a patient with p.Glu445*) carried the variant in non-mosaic state in blood and did not have ID or other relevant health issues.

### 3.2. Variants of Uncertain Significance and Possible Bi-Allelic Inheritance

Five missense variants are reported in the medical literature as variants of uncertain significance (VUS) [24], but many more are present in the ClinVar database (https://www.ncbi.nlm.nih.gov/clinvar/ accessed on 30 December 2023); we also describe here a single variant that may exert its pathogenic effect only in a homozygous state [26] (Table 2 and Figure 1).

Of the five heterozygous variants, two are in the HMG box domain, whereas three involve different regions of the protein. Again, both in silico and functional assessments were implemented for variant classification. These five variants are extremely rare (absent in gnomAD with the exception of p.Asp461Glu—one single allele reported). Regarding p.Lys132Arg (last position of the HMG box), it is interesting to note that Lys132 is conserved in SOXC proteins, but replaced by Arg in other SOX proteins including SRY. p.Ala316Ser, outside the HMG box, lies in an unstructured and functionally uncharacterised region, and Ala316 is poorly conserved. Two variants (p.Asp461Glu and p.Ser466Gly) targeted the SOX4 TAD. Neither of these five variants was found to be less (or more) abundant in transfected cells than wild-type SOX4, and all of them showed normal activity in the transactivation assay. Although they affect the HMG box, p.Ala112Gly and p.Lys132Arg could bind DNA.

The Ala112 position deserves a specific comment: four distinct variants have been described involving this residue. p.Ala112Pro and p.Ala112Val were observed in affected individuals and were shown to alter SOX4 function, whereas p.Ala112Gly (affected individual) and p.Ala112Thr (in gnomAD, thus presumably unaffected or mildly affected) showed normal activity on functional tests. This is an interesting proof of the limits of in silico assessment and the need to perform functional studies to ascertain variant pathogenicity.

Parents were available to be tested in 3/5 cases (p.Ala112Gly, p.Asp461Glu, and p.Ser466Gly) and the variant was always transmitted. The mother carrying the p.Asp461Glu variant had epilepsy, while the mother carrying the p.Ser466Gly had learning difficulties, childhood epilepsy, and a mood disorder. The mother with the p.Ala112Gly variant is also reported to be mildly affected.

On average, the clinical features associated with VUS were reported to be milder when compared with those defined as (likely) pathogenic, but the number of cases is still too limited. It is possible that these variants, although rare, are irrelevant with respect to the patients’ clinical features; otherwise, some/all of these variants may exert a pathogenic effect, possibly milder, and could be defined as “hypomorphic”. No functional tests are currently available to prove this hypothesis.

Further, a bi-allelic variant, an in-frame microdeletion [c.730_753del; p.(Ala244_Gly251del)], was reported in a single consanguineous family [26]. Two affected siblings had global developmental delay, moderate to severe ID, hypotonia, and mild facial dysmorphism. The parents were heterozygous for the p.(Ala244_Gly251del) variant and had normal IQ levels with no history of hypotonia or facial dysmorphism. This variant removes eight evolutionarily conserved amino acids within a functionally unknown SOX4 domain, suggesting that sequences outside the DNA-binding and transactivation domains could modulate SOX4 activity and thus undergo pathogenic alterations. Again, the same authors hypothesize that the p.(Ala244_Gly251del) variant might be a hypomorphic allele of *SOX4*.

Functional studies were not performed, but it is tempting to think about the possibility that SOX4 activity needs to reach a threshold not to compromise normal development. Either monoallelic loss of function variants or bi-allelic hypomorphic variants may thus be disease-causing. Bi-allelic inheritance is very rare in SOXopathies. A homozygous *SOX10* deletion was found to cause a severe form of four-limb arthrogryposis but, rather than representing hypomorphic variants, it was a co-dominant occurrence and parents were both affected by Waardenburg syndrome [27]. Hypotrichosis–lymphedema–telangiectasia is a very rare disorder caused by heterozygous loss of function variants affecting *SOX18*, but the first report also described unrelated individuals with homozygous missense variants (p.Ala104Pro and p.Trp95Arg) and healthy heterozygous parents [28]; functional studies were not performed. However, these may represent hypomorphic variants.

### 3.3. Co-Occurrence of Variants in Other Genes

The presence of pathogenic variations at two distinct loci that lead to the expression of two Mendelian conditions, which segregate independently, has been appreciated as a relatively frequent phenomenon after the introduction of large-scale sequencing studies. It is termed “dual molecular diagnosis”, and several reports have demonstrated that this scenario occurs in a percentage of approximately 5% among patients who received a molecular diagnosis [29]. The two diagnoses can be “distinct”, when the two conditions have different phenotypes, or “overlapping”, when at least some of the clinical features are common to the two disorders [30].

At least three *SOX4* patients have a second causative variant: (a) the proband carrying p.Trp69Gly has a distinct molecular diagnosis, a bleeding disorder caused by a stop–gain mutation (p.Gln106*) in *F11*; (b) the patient with p.Gly44Argfs*2 has a *TTN*-related cardiomyopathy caused by a stop–gain variant (p.Arg18985*), and, since heart defects are frequently associated with *SOX4*, the presence of cardiomyopathy may be considered an overlapping phenotype; (c) the proband carrying the p.Arg466Gly VUS has an associated autosomal dominant myopia caused by a frameshift variant (p.Arg179Valfs*224) in *SLC39A5*, and it is possible that a liability to developmental delay was worsened or unveiled by an associated disorder causing poor eye-sight.

Furthermore, two VUS of potential interest were reported, p.Pro639Arg in *PHF8* in the patient carrying the p.Tyr325* variant, and p.R1406H in *CHD4* in the patient carrying p.Ala112Gly. Both *PHF8* and *CHD4* encode chromatin remodellers, associated with human developmental disorders [31,32], which might interact with SOX4 and contribute to the clinical phenotype.

## 4. Copy Number Variants Involving SOX4

SOX4-containing 6p22.3 copy number variants (CNVs) are rare [33,34,35,36]. Genome coordinates and clinical information for seven individuals that we could retrieve from medical literature are summarized in Table 3. Six patients harbour deletions: four carry a similar deletion of approximately 2 Mb involving *SOX4* and three further genes (*MBOAT1*, *E2F3*, *CDKAL1*), and share a specific skeletal phenotype, mesomelic dysplasia “Savarirayan type”. A larger deletion (encompassing *ID4*) and a smaller one (encompassing *SOX4* only) were NOT associated with a skeletal dysplasia. It was hypothesized that the 2 Mb deletions bring limb enhancers into close proximity with *ID4* due to deletion of a topologically associated domain, resulting in the aberrant activation and misexpression of *ID4* in the limb bud, and causing a skeletal dysplasia [34]. Thus, this skeletal phenotype is seemingly unrelated to *SOX4* deletion. Three of these individuals are reported to suffer developmental delay, namely the ones with the largest and smallest deletion, and only one of the four patients with the 2 Mb deletion causing mesomelic dysplasia. Indeed, *SOX4* haploinsufficiency may be the cause of developmental delay in these patients, considering that the individual carrying the smallest deletion (involving *SOX4* only) had mild psychomotor delay (plus facial dysmorphism and cleft palate) [35] and the other genes involved in larger deletions have never been associated to abnormal brain development. On the other hand, two patients were reported to show normal psychomotor development with no speech delay (although they were small children) [34], and a third one [33] had congenital anomalies (heart defect and bilateral neurosensorial hearing loss due to malformed semicircular canals) but neurological development was not reported.

The role of somatic *SOX4* variation in tumour biology is beyond the scope of this work, but many research studies correlated increased expression of SOX4 with tumorigenesis and progression in several cancer types (reviewed in [37]). In this respect, it is well known that cancer and disorders of neurodevelopment share molecular pathways and mutations [38]. Somatic variations in genes that affect cell proliferation are often drivers of cancer, whereas germline pathogenic variants in the same genes, if they are expressed in an appropriate time-window of brain development, can lead to neurodevolpmental disorders [39,40]. It is intriguing that the only *SOX4* whole gene duplication reported to date was identified in a boy with dysmorphisms and paediatric cancer (neuroblastoma), although this CNV was of paternal inheritance [36]. Furthermore, the patient carrying the smallest deletion (389 kb, SOX4 only) had nephroblastoma, with a paradoxical increased tumour/normal kidney SOX4 expression ratio. 

Larger case series will be needed to understand whether constitutional CNVs involving *SOX4* indeed predispose carriers to develop paediatric cancer.

## 5. Phenotype

All affected individuals had speech delay, often in the context of global developmental delay; ID can be borderline to (rarely) severe, and hypotonia and behavioural concerns are very common. Facial dysmorphisms are reported in almost every patient, but full pictures are available in the medical literature for three patients only [16]; a few others are masked. A fairly specific facial phenotype was present in the first individuals reported in the literature: horizontal palpebral fissures, bulbous tip of the nose with anteverted nares, long philtrum, wide mouth with thin upper lip and cupid bow, posteriorly rotated ears [16]. A much larger dataset of facial images would be needed to define whether the facial phenotype is specific for this condition.

Associated congenital anomalies are also very common but heterogeneous. More than 50% of individuals with pathogenic variants have cardiovascular anomalies, especially ventricular septal defects, and it is worth mentioning that vascular anomalies are frequent and deserve a follow-up: three patients (all with a missense variant) had aortic aneurism, one (nonsense variant) had a double aortic arch, and another one (nonsense variant) had an anomalous coronary artery that required surgery. Also, a patient with a *SOX4* VUS had mild aortic dilatation (with a bicuspid aortic valve). Long-term follow-up studies will be needed to assess whether aortic aneurisms are more frequent in adult *SOX4* patients and whether they are progressive.

Otitis media and hearing loss were present in 5/19 patients, but one patient had dysplastic semicircular canals and a second one dysplastic ossicle. Hearing loss with “malformed” semicircular canals was also reported in a patient with CNV-deleting *SOX4* [33]. It is interesting that hearing loss due to hypoplasia/dysplasia of the semicircular canals is very rare, but is a typical sign of another SOXopathy (Waardenburg syndrome caused by *SOX10* mutations) [7] and may hint at a connection between these two conditions.

Poor eye-sight, palatal abnormalities, and genitourinary findings (especially hypospadias) are further common concerns.

Overall, the phenotype of patients with *SOX4* variants resembled patients with other neurodevelopmental diseases, including *SOX11*-related disorders [14], but some facial features in the context of a developmental delay, together with inner ear and/or vascular anomalies, should help to address the diagnosis and facilitate interpretation of exome/genome sequencing data.

## 6. Discussion

Heterozygous variants that abolish SOX4 transcriptional activity in vitro cause a human neurodevelopmental syndrome with associated dysmorphic features and inconstant congenital anomalies. The clinical phenotype is consistent with the fundamental roles of SOX4 in development and is likely to be associated to reduced expression of SOX4 target genes at critical points in embryonic and early postnatal development.

Initially, this syndrome was classified under the umbrella of CSS, similarly to *SOX11*-neurodevelopmental disorder, but both conditions lacked the most specific features of CSS (e.g., fifth-finger nail hypoplasia, corpus callosum agenesis, and hypertrichosis and hirsutism).

In OMIM (#618506), *SOX4*-related disorder is now classified as “intellectual developmental disorder with speech delay and dysmorphic facies” (CSS10 being listed as alternative title).

Many open questions remain, some of which are common to most recently identified rare genetic conditions, some specific to the role which is played by SOX4.

The first question is how to define *SOX4* VUS. Seventy such variants are reported in the ClinVar database, often in individuals reported to have ID. It is possible to perform functional studies to define the role of missense variants in the HMG box domain, but at the moment it is much harder to define missense and in-frame variants in other regions.

An aid in defining SOX4 VUS may come from DNA methylation profiling, which has been proven to be a useful biomarker for clinical diagnosis of many rare disorders, among them *SOX11*-related neurodevelopmental syndrome [15]. *SOX11* pathogenic variants were found to be associated with a distinct DNA methylation profile, which was also useful to separate this disorder from CSS. Given the redundant roles in development of SOX4 and SOX11, it would be extremely useful to test patients with *SOX4* pathogenic variants to understand if they have a distinct methylation profile. This would then turn into a useful tool to classify VUS.

An alternative approach that can help to define a clinical diagnosis and thus interpret atypical genetic variation could be computational facial analysis: thanks to the advancements in computer vision and machine learning, so called “next-generation phenotyping” approaches have been developed, such as GestaltMatcher, which could facilitate the diagnosis of facial image analysis [41]. This approach was proposed to be integrated into exome variant prioritization pipelines. Almost every patient with a *SOX4* pathogenic variant has facial dysmorphisms, and we expect that more variants will be identified and described, allowing for reverse phenotyping and definition of possibly typical facial features. GestaltMatcher may help to recognize facial similarities, determine a specific facial phenotype, and eventually be used to prioritize or reclassify uncertain variants [42]. Integrated approaches may be especially useful in a disorder where incomplete penetrance or mild clinical expression could hamper the detection of a causative variant, if it is transmitted by a seemingly healthy parent.

Other open issues are pathogenetic mechanism and genotype–phenotype correlations. Pathogenic missense changes are not associated with a less severe phenotype when compared with stop–gain pathogenic variants. Single-nucleotide pathogenic variants have been shown to abolish SOX4 transcriptional activity, so do pathogenic missense and nonsense variants primarily result in null alleles? Or do these variants maintain some functions with respect to true haploinsufficiency? On the other hand, as functional studies seem to indicate [24], some variants may be dominant-negative, precluding normal functioning of the wild-type allele. Clinical data on patients with CNVs causing SOX4 haploinsufficiency could help to clarify this point. Three out of five patients with large deletions involving *SOX4* were NOT reported to have a delay in neurological development, and, if these data were confirmed, they would indicate a milder phenotype caused by haploinsufficiency. As exemplified by *SALL1* mutations in Townes–Brockes syndrome, haploinsufficiency can exert a milder effect than recurrent truncating variants escaping NMD, which act through a dominant-negative mechanism [43].

The intrinsic function of the mutated SOX4 peptide (haploinsufficiency, total or partial lack of specific functions, dominant-negative effect) can of course determine clinical heterogeneity, but many other sources can be envisaged. Mosaicism is relatively frequent in other SOXopathies (e.g., Lamb–Shaffer syndrome) [44] and, whereas germinal mosaicism is important for recurrence risk evaluation, somatic mosaicism can be associated with milder phenotypes. Methodological advances have facilitated the detection of mosaicism [45]. Some de novo variants may be mosaics, and, ideally, tissues other than blood could be tested (e.g., DNA from skin biopsy), especially in patients who show phenotypes at the mild end of the clinical spectrum. At the same time, parents found to be carriers of *SOX4* variants (mosaic or not) should receive a clinical assessment wherever possible. 

Furthermore, “expansion” of the clinical spectrum of a known disorder may often hide a dual molecular diagnosis [46], as variants in other genes can modify the clinical phenotype. At least two patients with SOX4 pathogenic variants were reported to carry a causative variant in a distinct gene (F11, TTN); others had VUS that may contribute to the phenotype.

Some variants may be hypomorphic, exerting their effect when bi-allelic or on a multigenic/oligogenic background. To our knowledge, one single instance of *SOX4* bi-allelic variant has ever been reported [26], never of *SOX11*. Given the additive and redundant roles of SOX4 and SOX11 (and SOX12) in development, it is tempting to speculate that a source of clinical variability may be a synergistic heterozygosity for hypomorphic variants affecting these genes.

Treatment options and the best possible management for individuals with SOX mutation is the ultimate issue. No specific treatment is yet available, and if gene therapy could be an option for SOX4 or if SOX4 protein could be druggable remain unanswered questions. Gene supplementation therapies, favouring adeno-associated viruses (AAVs) as delivery vectors, are expanding the treatment options for neurological disorders [47], and an AAV gene therapy has been recently proven to be safe and effective in a neurodevelopmental disorder, aromatic L-amino acid decarboxylase deficiency [48]. To our knowledge, no clinical trials of gene therapy for SOXopathies are ongoing at the moment. A deeper understanding of SOX4 pathogenic mechanisms would be instrumental to plan precision therapies: briefly, a dominant-negative model could require silencing of the mutated peptide, whereas haploinsufficiency could be ameliorated via increasing expression of the wild-type allele.

However, individuals with SOX4 variants predicted to be damaging deserve prompt intervention for behavioural problems, which are common and can compromise development and social interactions in children and adults, although they often have only a mild degree of ID. Hearing and visual impairments are common and may compromise the intrinsic developmental potential of SOX4 patients. A close follow-up should be recommended for cardiovascular problems, with a possibly high risk of developing aortic aneurism, which should alert care-givers, especially at the time of adulthood transition. Finally, there are no reports of cancer in patients with single-nucleotide variants, but two carriers of 6p22.3 CNVs involving *SOX4* developed embryonal tumours; thus, we believe that individuals with such CNVs should be monitored during childhood.

In conclusion, these are exciting years for the definition of genetic disorders, including rare SOXopathies. Larger cohorts of patients will allow for a better clinical definition, personalized follow-up, and hopefully pave the way to targeted therapies.

## Figures and Tables

**Figure 1 genes-15-00158-f001:**
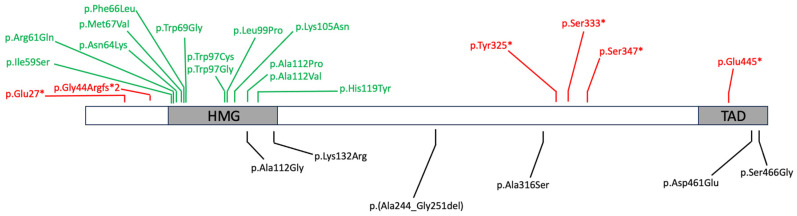
Representation of SOX4 protein with the two known domains (HMG: high-mobility group; TAD: transactivation domain). Known causative variants are reported above the protein (missense variants in green, stop–gain variants in red); variants of uncertain significance and the bi-allelic variant below.

**Table 1 genes-15-00158-t001:** Heterozygous *SOX4* causative variants and clinical phenotype.

Variant	Amino Acid Change	Type of Mutation	Inheritance	Hypotonia	ID/dev Delay	Behavior Anomalies	Seizures	Dysmorphic Features	Hands and Feet	Other Findings	Reference
c.79G>T	p.Glu27*	nonsense	De novo	No	Mild speech delay and learning disabilty	Yes	No	Yes	Nail hypoplasia Bilateral clinodactyly and hypoplasia of digits	Widely spaced nipples	[25]
c.130_133delGGCAinsCGCT	p.Gly44Argfs*2	frameshift	Mother (somatic mosaic, mildly affected)	Yes	Global dev delay	Yes	No	Yes	Bilateral 5th finger clinodactyly	Mild OSA, dysphagia, strabismus	[24]
c.176T>G	p.Ile59Ser	missense	De novo	NA	Mild ID and speech delay	Yes	No	Yes	Mild 5th finger clinodactyly and dysplastic 5th toenails	No	[16]
c.182G>A	p.Arg61Gln	missense	De novo	NA	Global dev delay	Yes	No	Yes	Relatively short fingers with short terminal phalanges and 5th finger clinodactyly	Chronic constipation, hypothyroidism, unilateral doubled kidney, WPW, aortic aneurism, scoliosis	[25]
c.192C>A	p.Asn64Lys	missense	Mildly affected father	No	Global dev delay	Yes	No	Yes	Bilateral 5th finger clinodactyly and brachydactyly of fingers and toes	Heart defect (VSD) and aortic aneurism	[24]
c.198C>A	p.Phe66Leu	missense	De novo	Yes	Global dev delay	NA	Yes	Yes	Bilateral 5th finger clinodactyly	Heart defect (VSD), feeding difficulties and constipation, strabismus	[16]
c.199A>G	p.Met67Val	missense	De novo	Yes (axial)	Global dev delay	Yes	No	Yes	No	Hearing loss, heart defect (VSD)	[24]
c.205T>G	p.Trp69Gly	missense	De novo	No	Global dev delay	Yes	No	Yes	Bilateral hypoplasia of 5th toe	Otitis media, supernumerary nipples, hypospadias, cleft palate	[24]
c.289T>G	p.Trp97Gly	missense	De novo	Yes	Global dev delay	Yes	Suspected	Yes	Nail hypoplasia	Hearing loss and preauricular tag, dysplastic ossicles with normal cochlea and semicircular canals, cleft palate	[24]
c.291G>C	p.Trp97Cys	missense	De novo	Yes	Global dev delay	Yes	No	Yes	NA	Otitis media, exotropia	[24]
c.296T>C	p.Leu99Pro	missense	De novo	Yes	Global dev delay	Yes	No	No	NA	Dysgenesis of corpus callosum and of semicircular canals, hearing loss	[24]
c.315G>T	p.Lys105Asn	missense	De novo	NA	Global dev delay	NA	NA	Yes	Mild 5th finger clinodactyly and mild camptodactyly	No	[16]
c.334G>C	p.Ala112Pro	missense	De novo	Yes	Global dev delay	NA	No	Yes	Bilateral 5th finger clinodactyly	Progressive cerebellar atrophy and spastic qudriparesis, laryngomalacia, microcephaly	[16]
c.335C>T	p.Ala112Val	missense	Uk	No	Speech delay	Yes	Yes	Yes	NA	Dysphagia, exotropia, aortic aneurism	[24]
c.355C>T	p.His119Tyr	missense	Unaffected mother	Yes	Speech delay and learning disability	Yes	Yes	Yes	Nail hypoplasia	No	[25]
c.975C>A	p.Tyr325*	nonsense	De novo	No	Global dev delay	Yes	No	No	Plump hands with small 5th fingers	NA	[24]
c.998C>A	p.Ser333*	nonsense	De novo	Yes	Global dev delay	Yes	No	Yes	Unilateral single transverse palmar crease	Heart defect (VSD), double aortic arch, short corpus callosum	[24]
c.1040C>A	p.Ser347*	nonsense	De novo	No	Mild ID and speech delay	Yes	No	Yes	No	Heart defect (ASD, VSD and abnormal pulmonary valve), chronic constipation	[24]
c.1333G>T	p.Glu445*	nonsense	Unaffected father	Yes	Global dev delay	Yes	No	Yes	Bilateral pes planus	Visual impairment, hypospadias, anomalous coronary artery	[24]

Dev: development; OSA: obstructive sleep apnea; NA: not assessed; ID: intellectual disability; WPW: Wolf-Parkinson-White; VSD: ventricular septal defect; Uk: unknown; ASD: atrial septal defect.

**Table 2 genes-15-00158-t002:** *SOX4* variants of uncertain significance, biallelic variant, and clinical phenotype.

Variant	Amino Acid Change	Type of Mutation	Inheritance	Hypotonia	ID/dev Delay	Behavior Anomalies	Seizures	Dysmorphic Features	Hands and Feet	Other Findings	Reference
c.335C>G	p.Ala112Gly	Missense	Mother (possibly affected)	NR	Mild ID	No	NR	Yes	NR	Hypospadias, cryptorchidism	[24]
c.395A>G	p.Lys132Arg	Missense	Uk	Yes	Global dev delay	Yes	No	Yes	No	No	[24]
c.730_753del	p.(Ala244_Gly251del)	In frame deletion	Het unaffected parents	Yes	Global dev delay	No	NR	Yes	No	No	[26]
c.946G>T	p.Ala316Ser	Missense	Uk	Yes	Global dev delay	No	No	Yes	No	OSA, bicuspid aortic valve with mild aortic dilatation, exotropia	[24]
c.1383C>G	p.Asp461Glu	Missense	Mother	Yes	Global dev delay	Yes	Yes	Yes	Bilateral short 5th toe	No	[24]
c.1396A>G	p.Ser466Gly	Missense	Mildly affected mother	Yes	Global dev delay	Yes	No	Yes	Slender fingers with broader thumbs and halluxes; prominent heels	No	[24]

NR: not reported; ID: intellectual disability; Uk: unknown; Dev: development; Het: heterozygous; OSA: obstructive sleep apnea.

**Table 3 genes-15-00158-t003:** 6p22.3 copy number variants including *SOX4* (in order of size) and clinical phenotype.

Coordinates (hg19)	Size	Copy Number	Inheritance	ID/dev Delay	Facial Dysmorphism	Skeletal Phenotype	Cancer	Other Findings	Reference
21,489,707–21,785,500	~290 Kb	Dup	Unaffected father	NR	Yes	No	Neuroblastoma	NR	[36]
21,390,094–21,778,878	~390 Kb	Del	De novo	Mild dev delay	Yes	No	Nephroblastoma	Posterior cleft palate	[35]
20,019,758–21,784,966	~1.8 Mb	Del	De novo	NR	Yes	Lower extremity hemimelia	No	Hypereosinophilic syndrome; hearing loss with malformed semicircular canals; heart defect	[33]
19,964,281–22,008,341	~2 Mb	Del	De novo	No	NR	Mesomelic dysplasia Savarirayan type	No	NR	[34]
19,974,194–22,013,061	~2 Mb	Del	Uk	Global dev delay	NR	Mesomelic dysplasia Savarirayan type	No	Bilateral supernumerary nipples, several Mongolian spots	[34]
19,849,280–21,604,600	~2 Mb	Del	De novo	No	NR	Mesomelic dysplasia Savarirayan type	No	No	[34]
19,153,386–21,698,497	2.5 Mb	Del	De novo	Global dev delay	NR	No	No	No	[34]

ID: intellectual disability; Del: deletion; Dev: development; NR: not reported; Uk: unknown; Dup: duplication.

## Data Availability

No new data were created or analyzed in this study. Data sharing is not applicable to this article.

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
