# Peer review of "Landscape of Constitutional SOX4 Variation in Human Disorders"

_genes, 2024, doi:10.3390/genes15020158_

Round 1

Reviewer 1 Report

Comments and Suggestions for Authors

The authors discuss the SOX group of genes, especially SOX4, and its association with genetic diseases. The gene group approach and its correlation with genetic diseases is quite interesting. The tables describing aspects of the variants were very good, and the article is a review with a peculiar approach. The authors provide an in-depth approach to SOX4 based on a literature review, and the conclusions are consistent with the evidence and arguments presented. They address the main question posed. The references are appropriate.

The paper is interesting and discusses a subject that is still not well understood.  Please include any additional comments on the tables and figures.

In table 1, 2 and 3, I suggest placing the acronyms below the table.

I suggest improving the quality of the image in Figure 1. The contrast between the letters and the background could be better, as well as the lines and variants described.

Author Response

Thank you for your support and comments. We improved the quality of Figure 1 and we moved the acronyms below the tables.

Reviewer 2 Report

Comments and Suggestions for Authors

This is a very well written review on the role of SOX4 in disease. Only a few minor comments-

1.     The review reads very long and elaborate and can benefit for editing and simplifying the text to cut down the length. Further, several parts of the text appear to sound like a summary of preliminary work that has been previously performed and lack novel idea generating concepts.

2.     It would be beneficial to readers if the authors, who seem like experts in the field provide their insights and expertise on this gene-disease association and its findings, rather than summarizing various findings. Ex. Can the authors speculate if missense changes are associated with less severe phenotypes than loss of function? If so, how is phenotype severity determined in this condition?

3.     Discussion is very long and repetitive, can be trimmed to be crisp.

4.     Table resolution is not very clear and needs to be improved.

Author Response

Thank you for your comments.

We tried to review and summarize previous findings, and indicate areas of future research. It is true that the discussion is rather long, but we did our best to avoid repetitions and expand ideas for further research.

Pathogenic missense changes are not associated with a less severe phenotype when compared with stop-gain pathogenic variants. Since SOX4 is a single-exon gene, stop-gain variants may be dominant-negative and exert a stronger pathogenic effect with respect to haploinsufficient variants. Thus, it is interesting to observe that whole-gene deletions (CNVs) seem to cause a milder phenotype… but larger case series will be needed to ascertain if this is a real effect. Also, missense variants defined as of “unknown significance” are associated, on average, with a less severe phenotype. But it is not clear yet if they are unrelated to the clinical phenotype or if they are hypomorphic variants.

We improved tables' resolution, although characters in table 1 are very small.

Reviewer 3 Report

Comments and Suggestions for Authors

This paper provides a review of the genetic and phenotypic landscape of SOX4 developmental disease. Overall the manuscript is excellently written and presented and provides a neat, well-organized overview of SOX4 in developmental disease. The Tables are clear and well developed and the Discussion extremely interesting.

A couple suggestions below:

The manuscript needs some English language editing. For example at line 307 “been proved” should be “been proven”. There are multiple such examples throughout the text.

It is not clear to me why you focus on SOX4 for in particular. It might be worth explaining why you zoom in particularly on this gene, after you’ve (very well) portrayed the landscape of SOX mutations in developmental disease.

In Table 1, can you specify that the mother with the mutation in the second row was somatic mosaic (even though it’s inferable since in the text you say all DNA was from peripheral blood)? This is of relevance since there is increasing knowledge that many neurodevelopmental disorders are of germline mosaic origin (even when thought of, for example, as de novo). Mosaicism will likely continue to emerge as an under-detected mode of inheritance as such, so specifying the type is critical.

Relatedly, your brief discussion of mosaicism in the Discussion warrants specification about whether you are referring to somatic or germline mosaicism, since is this highly relevant to the mechanism of disease but also of salience to clinicians searching for appropriate tissues to make genetic diagnoses from.

Figure 1 is interesting, but could you color the mutations according to mutation type (frameshift/nonsense/missense)? In addition, the cluster of mutations close to the TAD domain is of interest. Is this a highly conserved region? A TF binding site? Have you searched for any other associations? This would be of interest to any reader.

Your discussion of the Ala112 locus is interesting too. Can this be linked to any theories related to regions of hypermutability in the human (/vertebrate) genome?

In Table 3, is there a reason you switched categories from “ID/dev delay” (as in Tables 1 & 2) to “ID and neurological delay”? I’d recommend sticking to one category type, i.e. “ID/dev delay”, for consistency throughout.

Your discussion of the role of SOX4 in neurogenesis and tumorigenesis is very interesting. Since it is becoming increasingly clear that many NDD genes overlap with genes involved in process of cellular proliferation (and/or differentiation) (see https://www.sciencedirect.com/science/article/abs/pii/S0166223616000497) (e.g. NDD/cancer overlap https://www.nature.com/articles/s41525-023-00377-6), it may be worth adding 2-3 sentences to this effect to place your findings within the broader context of ongoing research on such genes. If you so wish, it could be wroth discussing the link to the spermatogonial selection hypothesis, of keen interest in the context of NDDs, and linked to cellular proliferation (see https://www.ncbi.nlm.nih.gov/pmc/articles/PMC4001324/).

In the Discussion, your brief mentioning of gene therapy could benefit from you referencing a couple of the recent (e.g. AAV based) breakthroughs in this space. This would be of interest to any researcher and/or clinician and strengthen your overall Discussion/contextualization of your work.

Comments on the Quality of English Language

Minor editing required.

Author Response

Thank you for your support, comments and useful suggestions.

The manuscript needs some English language editing. For example at line 307 “been proved” should be “been proven”. There are multiple such examples throughout the text.

English editing was improved.

It is not clear to me why you focus on SOX4 for in particular. It might be worth explaining why you zoom in particularly on this gene, after you’ve (very well) portrayed the landscape of SOX mutations in developmental disease.

A recent overview on general “SOXopathies” is available, as well as many reviews on specific SOX genes. We focused on SOX4 because disease-causing mutations have been recently defined but are growing rapidly in number.

In Table 1, can you specify that the mother with the mutation in the second row was somatic mosaic (even though it’s inferable since in the text you say all DNA was from peripheral blood)? This is of relevance since there is increasing knowledge that many neurodevelopmental disorders are of germline mosaic origin (even when thought of, for example, as de novo). Mosaicism will likely continue to emerge as an under-detected mode of inheritance as such, so specifying the type is critical.

We confirm (and report in the table) that this woman has somatic mosaicism.

Relatedly, your brief discussion of mosaicism in the Discussion warrants specification about whether you are referring to somatic or germline mosaicism, since is this highly relevant to the mechanism of disease but also of salience to clinicians searching for appropriate tissues to make genetic diagnoses from.

We expanded the discussion on mosaicism.

Figure 1 is interesting, but could you color the mutations according to mutation type (frameshift/nonsense/missense)? In addition, the cluster of mutations close to the TAD domain is of interest. Is this a highly conserved region? A TF binding site? Have you searched for any other associations? This would be of interest to any reader.

We changed the figure and added colour. Known functional domains are DNA-binding and TAD, there is no knowledge on functions of other regions, and conservation outside HMG-BOX and TAD is much lower.

Your discussion of the Ala112 locus is interesting too. Can this be linked to any theories related to regions of hypermutability in the human (/vertebrate) genome?

The two variants classified as pathogenetic hit two distinct nucleotides of this codon. In general, this region is not repetitive and there are no clear hints of a possible hypermutability at the moment.

In Table 3, is there a reason you switched categories from “ID/dev delay” (as in Tables 1 & 2) to “ID and neurological delay”? I’d recommend sticking to one category type, i.e. “ID/dev delay”, for consistency throughout.

We apologize and changed table 3 accordingly.

Your discussion of the role of SOX4 in neurogenesis and tumorigenesis is very interesting. Since it is becoming increasingly clear that many NDD genes overlap with genes involved in process of cellular proliferation (and/or differentiation) (see https://www.sciencedirect.com/science/article/abs/pii/S0166223616000497) (e.g. NDD/cancer overlap https://www.nature.com/articles/s41525-023-00377-6), it may be worth adding 2-3 sentences to this effect to place your findings within the broader context of ongoing research on such genes. If you so wish, it could be wroth discussing the link to the spermatogonial selection hypothesis, of keen interest in the context of NDDs, and linked to cellular proliferation (see https://www.ncbi.nlm.nih.gov/pmc/articles/PMC4001324/).

We added a few sentences on the role of genes in cancer and neurodevelopment and added some references.

In the Discussion, your brief mentioning of gene therapy could benefit from you referencing a couple of the recent (e.g. AAV based) breakthroughs in this space. This would be of interest to any researcher and/or clinician and strengthen your overall Discussion/contextualization of your work.

We discussed briefly gene therapy approaches for neurodevelopmental disorders.